# Quantitative Determination of Aflatoxin B_1_ in Maize and Feed by ELISA and Time-Resolved Fluorescent Immunoassay Based on Monoclonal Antibodies

**DOI:** 10.3390/foods13020319

**Published:** 2024-01-19

**Authors:** Shiyun Han, Yalin Yang, Ting Chen, Bijia Yang, Mingyue Ding, Hao Wen, Jiaxu Xiao, Guyue Cheng, Yanfei Tao, Haihong Hao, Dapeng Peng

**Affiliations:** State Key Laboratory of Agricultural Microbiology, National Reference Laboratory of Veterinary Drug Residues (HZAU) and MOA Key Laboratory for Detection of Veterinary Drug Residues, Huazhong Agricultural University, Wuhan 430070, China; hanshiyun2022@163.com (S.H.); yalin.yang@oost.com (Y.Y.); chent1217@gmail.com (T.C.); yangbijia2022@163.com (B.Y.); dmy00117@163.com (M.D.); w793653717@163.com (H.W.); xiao_jiaxu@163.com (J.X.); chengguyue@mail.hzau.edu.cn (G.C.); tyf@mail.hzau.edu.cn (Y.T.); haohaihong@mail.hzau.edu.cn (H.H.)

**Keywords:** aflatoxins B_1_, ELISA, immunochromatographic assay, TRFICA, maize

## Abstract

In this study, a highly sensitive monoclonal antibody (mAb) was developed for the detection of aflatoxin B_1_ (AFB_1_) in maize and feed. Additionally, indirect competitive enzyme-linked immunosorbent assay (ic-ELISA) and time-resolved fluorescence immunoassay assay (TRFICA) were established. Firstly, the hapten AFB_1_-CMO was synthesized and conjugated with carrier proteins to prepare the immunogen for mouse immunization. Subsequently, mAb was generated using the classical hybridoma technique. The lowest half-maximal inhibitory concentration (IC50) of ic-ELISA was 38.6 ng/kg with a linear range of 6.25–100 ng/kg. The limits of detections (LODs) were 6.58 ng/kg and 5.54 ng/kg in maize and feed, respectively, with the recoveries ranging from 72% to 94%. The TRFICA was developed with a significantly reduced detection time of only 21 min, from sample processing to reading. Additionally, the limits of detection (LODs) for maize and feed were determined to be 62.7 ng/kg and 121 ng/kg, respectively. The linear ranges were 100–4000 ng/kg, with the recoveries ranging from 90% to 98%. In conclusion, the development of AFB_1_ mAb and the establishment of ic-ELISA for high-throughput sample detection, as well as TRFICA for rapid detection presented robust tools for versatile AFB_1_ detection in different scenarios.

## 1. Introduction

Approximately 60–80% of global agricultural products are annually contaminated by mycotoxins, with aflatoxins (AFTs) playing a significant role in this proportion [1]. AFTs are secondary metabolites mainly produced by *Aspergillus flavus* and *A. parasiticus*, which are teratogenic and mutagenic [2]. Among them, aflatoxin B_1_ (AFB_1_) is considered a carcinogenic agent (group 1 carcinogens) due to its potent hepatocellular carcinoma (HCC) in humans, as well as its immunotoxic, mutagenic, and teratogenic properties in humans [3]. Cereals become contaminated by AFTs during their growth, harvesting, transportation, and storage [4]. Upon ingestion of these contaminated grains by animals, AFB_1_ accumulates within their bodies, subsequently leading to the contamination of animal-derived food products. The consumption of such contaminated foods poses a significant risk to human health, while the excretion of AFB_1_ metabolites in animal feces further contributes to environmental pollution [5]. In order to safeguard the well-being of both animals and humans, China, the FDA, and the European Union have established stringent regulations on maximum residue limits (MRLs). The MRLs of AFB_1_ in maize, peanut products, and other animal feeds are stipulated as 20 μg/kg by both China and the FDA. In contrast, the European Union sets the MRLs for AFB_1_ at 5 μg/kg in maize and rice, and at 2 μg/kg in grain and grain derivatives. Therefore, it is of great significance to establish a reliable and sensitive method for the detection of AFB_1_.

Currently, instrumental methods such as liquid chromatography coupled fluorescence detector [6] and liquid chromatography–tandem mass spectrometry [7] are standard methods for AFB_1_ detection, which have high sensitivity and stability. However, these instrumental methods have some disadvantages, such as expensive instruments, complicated sample processing procedures, and the need for professional technicians, rendering them unsuitable for swift on-site monitoring of large quantities of samples. Therefore, the development of a more rapid and convenient screening method is imperative. In comparison to instrumental techniques, immunoassays such as the indirect competitive enzyme-linked immunosorbent assay (ic-ELISA) and lateral flow immunoassay (LFIA) offer notable advantages including swiftness, high specificity, and exemption from intricate sample pretreatment requirements [8]. In recent years, an increasing number of commercial analytical kits based on ic-ELISA have been developed for the detection of AFB_1_ in various substrates [9,10]. Meanwhile, due to the simplicity and convenience of LFIA, many new marker materials have been developed in order to obtain lower detection limits of strips, among which the time-resolved fluorescence immunoassay assay (TRFICA) has emerged as a pivotal novel technique [11].

TRFICA is an innovative assay that integrates time-resolved fluorescence and immunoassay techniques, employing time-resolved fluorescent microspheres (TRFMs) as labeling agents. TRFMs are functional microspheres containing a large amount of bright lanthanide fluorescent materials, offering several distinct advantages: long fluorescence lifetime, large stokes shift, and effective elimination of various fluorescent background interference in the matrix. Therefore, compared with the classical gold nanoparticle-based strip assay (GNP-SA), the TRFICA method exhibits a detection sensitivity several-fold higher [11]. In recent years, TRFICA has emerged as a prominent research area in the quantitative analysis of mycotoxins and has been reported. Zhang et al. used a chromatographic time-resolved fluoroimmunoassay to determine the AFB_1_ in food and feed samples [12]. Tang et al. used an AIdnbs-TRFICA to determine dual mycotoxin in food and feed samples [13]. However, there is still a massive demand for the individual detection of AFB_1_ in maize and feed samples.

Therefore, this study developed a highly sensitive monoclonal antibody (mAb) for AFB_1_, enabling the establishment of an ic-ELISA method with improved sensitivity through optimization of the coating antigen. To further advance the rapid detection technology of AFB_1_, a TRFICA assay was established for detecting AFB_1_ in maize and feed samples, with the optimization of labeling conditions and strip materials.

## 2. Materials and Methods

### 2.1. Reagent and Equipment

Standard substances (aflatoxin B_1_, aflatoxin B_2_, aflatoxin G_1_, aflatoxin G_2_), N-hydroxysuccinimide, N′N-Dimethylformamide (DMF), tween-20, bovine serum albumin (BSA), N′N-dicyclohexylcarbodiimide (DCC), keyhole limpet (KLH), 1-ethyl-3-(3-dimethylaminopropyl)carbodiimide (EDC), dimethyl sulfoxide (DMSO), ovalbumin (OVA), O-carboxymethoxylamine hemihydrochloride (CMO), RPMI-1640, and horseradish peroxidase-labeled goat anti-mouse immunoglobulin G (HRP-IgG) were supplied by Thermo Fisher Scientific (Waltham, MA, USA), while hypoxanthine aminopterin thymidine (HAT) 3,3,5,5-tetramethylbenzidine (TMB), polyvinyl pyrrolidone (PVP), complete Freund’s adjuvants, incomplete Freund’s adjuvants, hypoxanthine thymidine (HT), albumin human serum (HSA), and polyethylene glycol 1500 (PEG) were purchased from Sigma-Aldrich (USA). Time-resolved fluorescent microspheres with 200 nm were purchased from Shanghai Huge (Shanghai, China). All of the other chemicals solvents were analytical grade. SP2/0 mouse plasmacytoma was provided by the National Reference Laboratory of Veterinary Drug Residues (HZAU), Wuhan, China.

XYZ3010 Dispensing Platform and CM3020 Guillotine Cutter was from JN Bio, HM30305 Batch Laminator from Kinbio (Shanghai, China). The high-speed freezing centrifuge (H1850R) was from Xiangyi (Changsha, China). Nitrocellulose membranes, sample pads, and absorbent pads were purchased from JY Bio (Shanghai, China). All of these were used to make TRFICA strips. The TRFICA results were read by a TRFICA strip reader FIC-S100 from Femdetection Bio-Tech (Shanghai, China).

### 2.2. Synthesis of Hapten and Antigen

#### 2.2.1. Synthesis of Hapten AFB_1_-CMO

The synthesis of AFB_1_-CMO hapten was performed according to a previously described method with some modifications [14]. Briefly, as shown in Appendix A, AFB_1_ (2 mg) was dissolved in DMF solution, followed by the addition of Na_2_CO_3_ (5 mg) and 25 mg CMO (25 mg). The reaction mixture was agitated at room temperature for 12 h, then was evaporated to dryness under N_2_ and redissolved in HCl (2 mL, 0.05 mol/L), and ethylacetate (2 mL) was used to extract the AFB_1_-CMO hapten. The obtained liquid was blown dry at N_2_ to obtain hapten AFB_1_-CMO and stored at 4 °C.

#### 2.2.2. Synthesis of Antigen

Immunogens (AFB_1_-DCC-KLH and AFB_1_-EDC-HSA) and coating antigens (AFB_1_-EDC-BSA and AFB_1_-EDC-OVA) were synthesized by using the active ester method [15]. Briefly, AFB_1_-CMO was dissolved in DMF (500 μL) and named liquid A; then, 10 mg DCC and 5 mg NHS were added into liquid A (200 μL) and stirred continuously at room temperature for 12 h and named liquid B; 2 mg KLH was dissolved in 5 mL PBS and named liquid C (5 mL); then, slowly added liquid B into liquid C and reacted in the ice bath. Finally, PBS was dialyzed at 4 °C for 72 h to remove unreacted AFB_1_-CMO, and then the supernatant was stored at −20 °C. The synthesis of AFB_1_-EDC-HSA, AFB_1_-EDC-BSA, and AFB_1_-EDC-OVA followed a similar procedure as that described for the synthesis of AFB_1_-DCC-KLH, except that DCC was substituted by EDC and KLH was substituted by HSA, BSA, or OVA.

The conjugate synthesis was verified and the hapten/protein ratios were estimated using a UV–visible spectrophotometer. The number of hapten residues conjugated to the carrier molecules was estimated using the ultraviolet absorbance spectra of the haptens, carrier proteins, and conjugates as follows: [ε (_conjugation_) − ε (_protein_)]/ε (_hapten_), where ε is the absorbance coefficient of the analytes.

### 2.3. Synthesis of Monoclonal Antibody

Six BALB/c mice (female, 6–8 weeks) were inoculated with an equal volume of Freund’s adjuvant (500 µL) and AFB_1_-DCC-KLH (500 µL) or AFB_1_-EDC-HSA (500 µL). After the first immunization, a second immunization is given three weeks apart, followed by a third immunization two weeks apart. The first injection employed a completed adjuvant, while the second and third injections utilized an incomplete adjuvant. The immunogen emulsion was subcutaneously administered to multiple sites on the dorsal surface of the mice. Collecting blood and the antiserum titer was determined by the ic-ELISA method. Then, mice exhibiting optimal dose–response curves were selected for fusion. The spleen cells from immunized mice were fused with sp2/0 myeloma cells at a ratio of 10:1~5:1, following the previous steps [16]. To ensure the monoclonal origin, hybridomas displaying positive reactions were subcloned twice through limiting dilution. Subsequently, ascites were induced in mice by intraperitoneal injection of the hybridoma, followed by identification of antibody subclasses and classes secreted using a mouse monoclonal isotyping kit (Proteintech Group, Inc., Chicago, IL, USA). The cell culture supernatant was assessed using ic-ELISA and mAbs exhibiting the lowest half-maximal inhibitory concentration (IC_50_) against AFB_1_ were selected for further investigation.

### 2.4. Establishment of the ic-ELISA

Coating antigen (100 μL, 0.54 μg/mL AFB_1_-EDC-OVA in carbonate–bicarbonate buffer, pH 9.6) was added to microplates and incubated at 4 °C overnight. After washing thrice with PBST (0.05% Tween 20 in 0.1 M phosphate-buffered saline, pH 7.4), microplates were blocked with blocking agents (250 μL, 1% OVA in 0.1 M phosphate-buffered saline (PBS), pH 7.4) for 2 h at 37 °C. After another washing step, 50 μL sample or a 50 μL _AFB1_ solution (diluted with 0.1 M PBS to 0, 6.25, 12.5, 25, 50, and 100 ng/L, respectively) was added; then, 50 μL diluted antibody was added and incubated at 37 °C for 40 min. Following a washing step, HRP-labeled goat anti-mouse IgG (diluted with 0.1 M PBS at 1:10,000) was added to each well (100 μL) and incubated at 37 °C for 40 min. After another washing step, TMB substrate solution (100 μL) was added; 15 min later, the enzymatic reaction was terminated with 1 M H_2_SO_4_ (50 μL). Finally, the absorbance was measured at 450 nm wavelength.

The checkerboard method was employed to determine the optimal concentration of coating antigen and antibody dilution. Different levels of AFB_1_ (0, 6.25, 12.5, 25, 50, and 100 ng/L) standard solutions were detected with the ic-ELISA, and a standard curve was fitted to an equation. Cross-reactivity (CR) was assessed by determining half-inhibitory concentration (IC_50_) values of AFB_1_, AFB_2_, AFG_1_, and AFG_2_. The CR values were calculated as follows: CR = (IC_50_ of AFB_1_)/(IC_50_ of competitor) × 100%.

### 2.5. Preparation of TRFICA and Procedure

The TRFICA system comprises a sample pad, a conjugate pad, an absorbent pad, an NC membrane, and a plastic adhesive backing card. The strip operates on a competitive scheme due to the low molecular nature of the target analyte. The AFB_1_-BSA and goat anti-mouse IgG were diluted to the proper concentration and fixed on the NC membrane at the speed of 0.8 μL/cm as the detection line (T line) and control line (C line) with the width was 5 mm and then the coated NC membranes were dried at 37 °C overnight. The prepared TRFM-mAb immunoprobe solution (see Appendix A for details) was uniformly dispersed on the bonding pad at a rate of 2.5 μL/cm and dried at 37 °C. Lastly, the conjugate pad, sample pad, NC membrane, and absorbent pad were laminated and coated onto the backing pad. The assembled backing was then cut lengthwise to a width of 3.7 mm and installed in the cartridge for composition testing.

The specific detection steps are as follows: dilute the treated sample solution to an appropriate concentration, absorb 100 μL of the solution to be tested and add it to the sample pad of the test strip, followed by incubation at 37 °C for 6 min, and then qualitatively evaluate the results under an ultraviolet lamp. The fluorescence intensity of the T line and C line of paper was recorded via strip reader for quantitative analysis.

The type of binding pad (glass fiber, polyester fiber), the type of NC membrane (BB-100, A × C 100, CN 120, PALL 190, PALL 120, PALL 170), the type of sample pad (GF2-S, Fusion 6, MA0120), the concentration of AFB_1_-BSA on the T line (0.3, 0.5 mg/mL), the concentration of goat anti-mouse IgG on C line (0.2, 0.3 mg/mL), and the detection time were optimized. In these experiments, 100 μL PBS buffer and AFB_1_ standard solution (2 μg/L) were added to the sample for detection, and each group was repeated 3 times.

AFB_1_ fluorescence strips were prepared according to the above optimal conditions, and different levels of AFB_1_ (0, 100, 250, 500, 1000, 2000, 4000 ng/L) standard solutions were detected with the same batch of strips, and each concentration was detected five times. After the reaction, the fluorescence intensity of C and T lines were read and T/C values were calculated. The B/B0 value was determined using the formula B/B0 = T/C value (positive) ÷ T/C value (negative). With the logarithm value of AFB_1_ standard solution concentration as the horizontal coordinate and B/B0 as the vertical coordinate, the standard curve is drawn to obtain the regression equation and correlation coefficient.

AFB_1_ standard solution of 1 μg/L was used as a control, and ZEN, OTA, T-2, and DON standard solutions of 100 μg/L were respectively detected with the same batch of strip. Each concentration was detected three times.

### 2.6. Validation of the ic-ELISA and Time-Resolved Fluorescent Immunochromatographic Assay (TRFICA)

The sensitivity, accuracy, precision, practicality, and stability of ic-ELISA and TRFICA were evaluated by using the testing limit of detection (LOD), limit of quantitation (LOQ), coefficient of variation (CV), and recovery rate.

Maize and feed samples were provided by the Breeding Pig Testing Center and Veterinary Hospital (Huazhong Agricultural University, Wuhan, China). All samples were ground and sieved through a 40-mesh screen. Samples of 2 g were placed into a 50 mL centrifuge tube and 10 mL of methanol-water (7:3, *v*/*v*) was added [5]. All samples were then swirled for 5 min, filtered with Whatman No. 1 filter paper, and 1 mL of filtrate was diluted into 10 mL (PBS 0.01 mol/L, pH 7.4). After all steps are completed, the sample is filtered 2 times and then used for testing.

The OD and T/C values of 20 blank samples were determined according to the method of establishing a standard curve. The OD value or T/C value is substituted into the standard curve regression equation, the corresponding concentration is calculated, and the mean ($\bar{X}$) and standard deviation (SD) of the 20 concentration values are obtained. The detection limit and quantitative limit were calculated according to the formula limit of detection (LOD) = $\bar{X}$ + 3 × SD and formula limit of quantification (LOQ) = $\bar{X}$ + 10 × SD, respectively, to evaluate the sensitivity of the method. AFB_1_ concentrations of 1 × LOQ, 2 × LOQ, and 4 × LOQ were added to maize and feed, and the AFB_1_ concentration was determined via the ic-ELISA method. According to the Chinese feed standard (GB 13078-2017 [17]), AFB_1_ concentrations of 1 × MRL, 2 × MRL, and 4 × MRL were added to maize and feed, and the concentration of AFB_1_ was determined with the TRFICA. Five parallel samples were set up for each sample concentration, and three batches were repeated to calculate the recovery rate and coefficient of variation.

In order to verify the reliability of the developed method, LC-MS/MS was carried out to compare. The blank maize samples were detected by employing LC-MS/MS, ic-ELISA, and TRFICA simultaneously. According to a previous method [18], HPLC-MS/MS analysis was performed in positive ion mode (ESI+), and separation was achieved by using the C18 analytical column (150 mm × 2.1 mm × 3.5 µm). Further, the separation condition of HPLC-MS/MS was performed by using gradient elution with 1% formic acid as mobile phase A and methanol/acetonitrile (1:1) as mobile phase B. And injection volume was 20 µL, with 0.3 mL/min for the flow rate. The blank corn samples were supplemented with appropriate amounts of AFB_1_ (50, 100, 150, 200, 250, and 300 ng/kg and 0.1, 0.25, 0.5, 1, 2, and 3 μg/kg) individually. LC-MS/MS, ic-ELISA, and TRFICA were employed for simultaneous detection. Each sample was repeated twice, and the correlation between LC-MS/MS, ic-ELISA, and TRFICA was assessed using linear regression.

## 3. Results and Discussion

### 3.1. Characterization of Haptens and Antigens

The hapten AFB_1_-CMO was synthesized by conjugating AFB_1_ with CMO, as confirmed by LC/MS-IT TOF mass spectrometry analysis (Appendix A). As shown in Figure 1, the ultraviolet–visible (UV) absorption spectra of immunogen AFB_1_-DCC-KLH (λ_277_ nm) and AFB_1_-EDC-HSA (λ_275_ nm) and coating antigens AFB_1_-EDC-OVA (λ_276_ nm) and AFB_1_-EDC-BSA (λ_275_ nm) are different from those of KLH (λ_279_ nm), OVA (λ_279_ nm), BSA (λ_276_ nm), and AFB_1_ (λ_359_ nm), which indicated that all conjugates were successfully synthesized. In addition, the incorporation rates of AFB_1_-DCC-KLH, AFB_1_-EDC-HSA, AFB_1_-EDC-OVA, and AFB_1_-EDC-BSA were 4.4, 4.9, 11.8, and 5.6, respectively.

### 3.2. Identification of Antiserum and Monoclonal Antibody

Since different doses and inoculation intervals significantly affect the immune response, it is necessary to design a rational immunization regimen to obtain highly sensitive and specific antibodies. The ic-ELSIA results of mouse tail vein serum collected on day 7 after the third immunization indicated that all mice died at a dose of 100 μg, while serum titers and specificity were low at a dose of 50 μg (Appendix A). The dose of immunogen AFB_1_-EDC-HSA only affects the titer of mice, but no immune response can be generated regardless of the dose after immunization. Therefore, mice with immunogen AFB_1_-DCC-KLH and an immune dose of 80 μg were selected for cell fusion.

The immunized mouse spleen cells were fused with myeloma cells (sp2/0). The splenic cells of mouse 1 and mouse 3 in the immunogen AFB_1_-DCC-KLH (80 μg) group were fused with sp2/0, and hybridomas with secret antibodies meeting the detection requirements were screened. Two hybridoma cells 3B9 and 1B6 were rescreened and subcloned two times. As shown in Appendix A, the supernatant of cell 3B9 showed a higher OD value and greater inhibition rate than the supernatant of cell 1B6. In addition, 3B9 mAb is more selective to AFB_1_ than 1B6 mAb. Finally, cell 3B9 was selected for subcloning (The identification of antibody subclasses and light chains of 3B9 is shown in Appendix A).

### 3.3. ic-ELISA Performance Test

The primary selection of ic-ELISA is very important, and the sensitivity of the antibody can be greatly improved by using the heterogenic coating. Meanwhile, the coating concentration and antibody dilution were optimized, and AFB_1_-EDC-BSA was finally selected as the coating source, the optimal coating concentration was 1.08 μg/mL, and the optimal antibody dilution was 1:1500. More details were in Appendix A.

AFB_1_ (1 mg/mL) was diluted with PBS to six concentrations of 0, 6.25, 12.5, 25, 50, and 100 ng/L, ic-ELISA was performed, and the data were recorded. In order to obtain a higher correlation coefficient (R^2^), linear regression analysis was chosen. As shown in Figure 2a, the standard curve is drawn with lg[C(AFB_1_)] as the horizontal coordinate and B/B0 as the vertical coordinate, and the regression equation is obtained: y = −0.6339x + 1.5056 (n = 5); the R^2^ is 0.9939. The IC_50_ calculated by the logistic function is 38.58 ng/L and the linear range is 0~100 ng/L. According to this equation, the LOD of the developed ic-ELISA is 8.61 ng/L. As shown in Appendix A, CR of AFB_1_, AFB_2_, AFG_1_, and AFG_2_ were 100%, 7.31%, 3.04%, and 2.64%, respectively. The results showed that the antibody had high specificity.

### 3.4. Time-Resolved Fluorescent Immunochromatographic Assay (TRFICA)

As shown in Figure 3, the target analyte competes with AFB_1_-BSA to bind to a limited number of probes, and the analyte preferentially binds to the probe antibody to inhibit antigen binding. If the sample is negative, the fluorescence probe binds to both T and C lines, resulting in simultaneous fluorescence signals on both T and C lines. In contrast, if the sample is positive for AFB_1_, the fluorescence probe selectively binds to AFB_1_ molecules present in the sample, thereby hindering its binding with the T-line while leaving the C-line unaffected. Consequently, this preferential binding leads to a reduction or even absence of the fluorescence signal on the T-line while maintaining an unchanged fluorescence signal on the C-line. The fluorescence intensity of the T line decreases with the increase in AFB_1_ content, but the fluorescence intensity of the C line has no significant change. The final signal analysis which is inversely proportional to the concentration of the analyte and is based on the ratio of T to C lines (T/C) to eliminate the change of the strip and matrix effect. Therefore, a standard curve can be obtained by plotting the T/C to the AFB_1_ content concentration.

However, before establishing the standard curve, some parameters in the analysis process need to be optimized. We focused on optimizing the following parameters: the concentration of coating antigen and goat anti-mouse IgG was 0.3 mg/mL; the bonding pad was made of polyester fiber; the NC membranes were PALL 170 type; the sample pad model was MA0120; the best detection time was 6 min. More details have been provided in the Appendix A.

As shown in Figure 2b, the AFB_1_ standard solution concentration is taken as the horizontal coordinate and B/B0 is taken as the vertical coordinate to draw the standard curve. In order to obtain a higher R^2^, a four-parameter fitting curve analysis was chosen. The regression equation is y = (1.04 + 0.0437)/[1 + (x/0.748)^^1.01^] − 0.0437 (n = 5), and the correlation coefficient (R^2^) is 0.9979, which has a well-fitting linear relationship in the range of 0.1–4 μg/L. The calculated LOD is 60 ng/L and the LOQ is 87 ng/L.

The specificity of the fluorescent strip was evaluated using four mycotoxins (ZEN, OTA, T-2, and DON), and the corresponding results are presented in Table 1. The fluorescent strip prepared in this study exhibited inhibitory effects solely on AFB_1_; even at the level of 100 μg/L, the strip had no competitive inhibitory effect on ZEN, OTA, T-2, and DON, indicating that the fluorescence strip had high specificity.

### 3.5. Method Validation

To validate the newly developed ic-ELISA and TRFICA, the accuracy, precision, and reproducibility were evaluated by using spike-and-recovery analysis.

Three concentrations of AFB_1_ were added to the blank samples of maize and feed, and then the samples were treated and detected by ic-ELISA and TRFICA. As shown in Table 2, the recoveries of AFB_1_ added by ic-ELISA and TRFICA were 73 ± 3–94 ± 8% and 90 ± 0.1–98 ± 0.1%, respectively. The LOD and LOQ of ic-ELISA in maize samples were 6.58 ng/kg and 10.5 ng/kg, and the LOD and LOQ of ic-ELISA in feed samples were 5.54 ng/kg and 8.36 ng/kg. The LOD and LOQ of TRFICA in maize samples were 62.7 ng/kg and 102 ng/kg, and the LOD and LOQ of TRFICA in feed samples were 121 ng/kg and 201 ng/kg.

The blank maize samples were detected with LC-MS/MS, ic-ELISA, and TRFICA, respectively, and the results are shown in Figure 4. The correlation coefficient between LC-MS/MS and ic-ELISA is R^2^ = 0.9985. The correlation coefficient between ic-ELISA and TRFICA is R^2^ = 0.9889. The results of the two methods have good correlation, and they show that the two methods established in this experiment are effective and reliable and can be used as detection tools for AFB_1_ in real samples.

### 3.6. Comparison of Published Detection Methods for Major AFB_1_

The ELISA and LFIA for AFB_1_ detection in the past decade were summarized. In comparison to these methods, we evaluated the ic-ELISA and TRFICA techniques established in this study. As shown in Table 3, the total sample analysis time of the TRFICA method is much less than that of the ic-ELISA method in the absence of complicated sample pretreatment procedures and sophisticated instruments operated by professionals. Additionally, it demonstrates higher sensitivity than classical GNP-SA and shows the characteristics of simple operation and flexible application, and it is suitable for simultaneous detection of a single sample and on-site rapid detection. Although the ic-ELISA method necessitates a lengthy incubation time, its ability to simultaneously detect multiple samples on each 96-well polystyrene microdrop plate makes it highly suitable for high-throughput analysis of numerous samples. The TRFCA-LFIA we established exhibits superior detection capability, heightened sensitivity, reduced detection time, and enhanced recovery rate in maize and feed samples compared to previously reported multiple LFIA methods. As an economical, lightweight, and portable tool, LFIA has dominated the market for rapid detection. Currently available LFIA assays still rely on colloidal gold particles as a labeling material. However, the increasing demand for enhanced sensitivity presents challenges to the existing framework of this approach. The sensitivity of the paper strip was enhanced in this study through optimization of antibody performance and utilization of fluorescent microsphere labeling materials. Despite these improvements, the cost associated with time-resolved fluorescence immunoassay (TRFICA) remains higher compared to gold nanoparticle-based sandwich assay (GNP-SA). Consequently, the high expenses of materials and detection tools continue to serve as a significant barrier to its widespread adoption in the market.

## 4. Conclusions

In this study, a highly sensitive mAb 3B9 was prepared for the detection of AFB_1_ and utilized as the foundation for both ic-ELISA and TRFICA assays. The IC_50_ of AFB_1_ was 38.58 ng/kg with ic-ELISA, while the lowest detection limit of 20 samples was 5.54 ng/kg. The limits of detection for TRFICA were 60 ng/kg and 121 ng/kg in maize and feed, respectively, with a rapid detection time of only 6 min. These results exhibited excellent agreement with those obtained using LC-MS/MS. Compared with the two methods established in this study, both approaches demonstrate high-throughput quantitative analysis capabilities for AFB_1_. However, ic-ELISA exhibits superior sensitivity, while TRFICA offers a simpler and less time-consuming operational procedure. Two methods can meet the MRL of 20 μg/kg in maize and feed formulated by China and the CAC (Codex Alimentarius Commission), and the detection results of the two methods are consistent when detecting the same samples. To further enhance the sensitivity of the paper strip, we propose that the results could be analyzed utilizing surface-enhanced Raman scattering technique or substituting monoclonal antibodies with anti-idiotypic nanobodies. The ic-ELISA method can detect a large number of samples and TRFICA can realize individual and rapid detection of samples. This study provided efficient methods that can be used to detect AFB_1_ in different scenarios.

## Figures and Tables

**Figure 1 foods-13-00319-f001:**
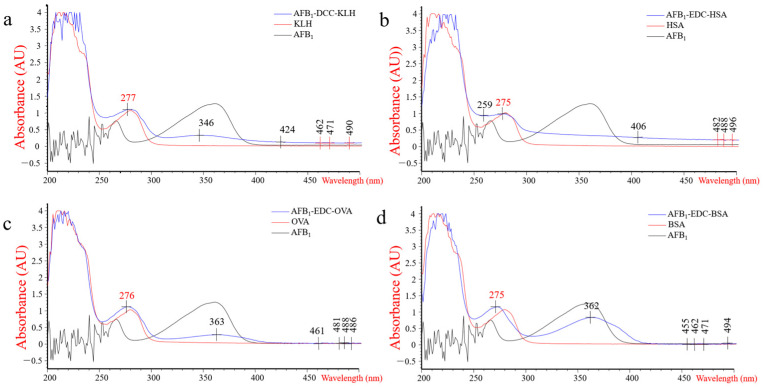
The UV absorption spectra of immunogen and coating antigens. (**a**) UV spectrum of AFB_1_-DCC-KLH, KLH, and AFB_1_. (**b**) UV spectrum of AFB_1_-EDC-HSA, HSA, and AFB_1_. (**c**) UV spectrum of AFB_1_-EDC-OVA, OVA, and AFB_1_. (**d**) UV spectrum of AFB_1_-EDC-BSA, BSA, and AFB_1_.

**Figure 2 foods-13-00319-f002:**
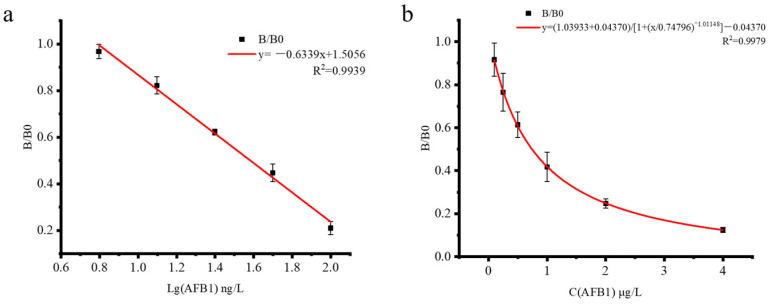
Standard curve of two methods. (**a**) Standard curve of ic-ELISA for AFB_1_. (**b**) Standard curve for fluorescence quantitative analysis of AFB_1_.

**Figure 3 foods-13-00319-f003:**
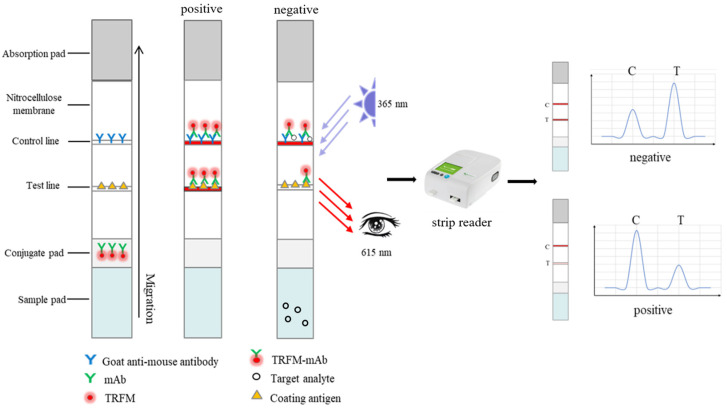
Principle and procedure of the TRFICA strip.

**Figure 4 foods-13-00319-f004:**
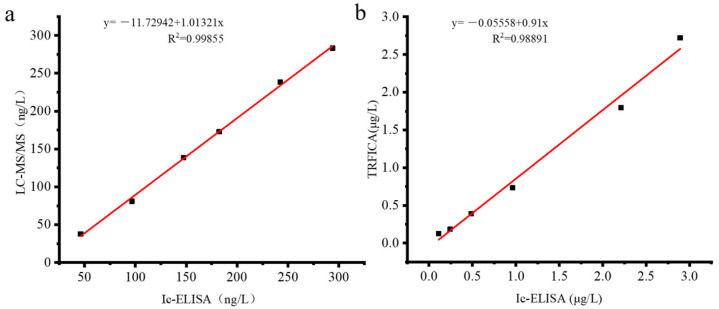
The linear regression analysis. (**a**) Correlation between TRFICA and ic-ELISA. (**b**) Correlation between ic-ELISA and LC-MS/MS.

**Table 1 foods-13-00319-t001:** Specificity of TRFICA.

Mycotoxins	T/C	CV%
AFB_1_	0.2849	100
ZEN	2.163	13.2
OTA	2.066	13.8
T-2	2.026	14.1
DON	2.303	12.4

**Table 2 foods-13-00319-t002:** Validation and comparison results of AFB_1_ spiked in maize and feed samples by ic-ELISA and TRFICA.

Method	Sample	Spiked (ng/kg)	LOD (ng/kg)	LOQ (ng/kg)	Intra-CV (%, n = 5)	Recovery (%)	Inter-CV (%, n = 15)
Ic-ELISA	Maize	11	6.58	10.5	5.53	83 ± 3	3.34
		22			4.59	94 ± 8	8.19
		44			5.82	75 ± 6	7.96
	Feed	15	5.54	8.36	5.12	84 ± 4	4.56
		30			5.76	80 ± 3	4.25
		60			5.10	72 ± 3	3.89
TRFICA	Maize	250	62.7	102	7.79	94 ± 0.1	13.6
		500			6.04	95 ± 0.09	9.05
		1000			3.18	93 ± 0.1	14.3
	Feed	500	121	201	7.10	90 ± 0.1	10.3
		1000			3.66	96 ± 0.1	11.2
		2000			7.64	98 ± 0.1	11.6

**Table 3 foods-13-00319-t003:** Comparison of the published method for the detection of AFB_1_.

Experimental Method	Marking Material	Target Analyte	LOD (μg/kg)	Sample Preparation	Total Analysis Time (min)	Reference
ELISA	-	AFB_1_	4.36	feed	15	[19]
Ic-ELISA	-	AFB_1_	0.008–0.020	animal-derived medicine	135	[20]
MBs-ic-ELISA	-	AFB_1_	0.0013	buffer solution	85	[21]
Ic-ELISA	-	AFB_1_	0.0386	feed and maize	85	This study
GICA	gold nanoparticles	AFB_1_	10	animal feeds or food	10	[22]
ICS	gold nanoparticles	AFB_1_, ZEN, and T2	0.5, 5.0, and 5.0	medicinal and edible food	10	[23]
LFIA	QDs	DON, ZEN, and T2/HT2	1000, 80, and 80	barley	15	[24]
LFIA	NU66@QD	AFB_1_, FB_1_, DON, T-2, and ZEN	0.04, 0.28, 0.25, 0.09, and 0.08	cereals and feed	8	[25]
LFIA	fluorescent microsphere	AFB_1_	3.4	distillers’ grains	15	[26]
TRFICA	TRFMs nanospheres	AFB_1_ + B_2_ + G_1_ + G2	0.16	Feed	12	[27]
TRFICA	TRFMs nanospheres	AFB_1_, ZEN	0.05 and 0.07	maize	8	[13]
TRFICA	TRFMs	AFB_1_ and ZEN	0.60 and 0.40	Chinese herbal medicines	15	[28]
TRFICA	TRFMs nanoparticle	AFTs, carbaryl, and carbofuran	0.03, 0.02, and 60.2	maize	10	[29]
TRFICA	TRFMs	AFB_1_, ZEN, DON, T-2, FB1	2.5, 0.5, 0.5, 2.5, and 0.5	maize, wheat, bran	8	[30]
TRFICA	TRFMs	AFB_1_	0.0627 and 0.121	maize and feed	6	This study

## Data Availability

Data is contained within the article and Appendix A.

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
