# Peer review of "Quantitative Determination of Aflatoxin B1 in Maize and Feed by ELISA and Time-Resolved Fluorescent Immunoassay Based on Monoclonal Antibodies"

_foods, 2024, doi:10.3390/foods13020319_

Round 1

Reviewer 1 Report

Comments and Suggestions for Authors

1. The authors should use full word (Aflatoxin B1) at the title of the manuscript.

2. The authors described that Chromatography is the gold standard method. Did the authors compare the results with the gold standard method? If not , please briefly add the justification for not using the gold standard method in this study.

3. The authors described that the newly developed method can reduce the time up to 20 minutes and plea se also describer the time  for routine analysis. At some reference the total detection time is 15 minutes. So, can the authors make conclusion reducing the time?

4. For ic ELSA sysem, the standard curve showed linearity but the fluorescent method, it is non-linearity. Please discuss briefly about the benefits and disadvantages at the revised manuscript.

5. Regarding the keywords, monoclonal Ab is not important and the author should add "maize" at the revised one.

Author Response

回应:非常感谢您对我们稿件的鼓励和有益的评论。您的支持真正激励着我们继续前进。您提出的建议和问题对文章的改进非常有价值。我们仔细地逐行修改了它,以红色突出显示更改,以方便您在审查时使用。我们再次衷心感谢您的时间和精力。

Reviewer 2 Report

Comments and Suggestions for Authors

The work describes quantitative determination of aflatoxin B1 in maize and feed using indirect elisa and time resolved fluorescence immunoassay. The authors thoroughly described the synthesis of the various haptens, antigens and the production of monoclonal antibodies for the  execution of the final assay procedures. Despite the hard work put into the work the majority of figures were presented with poor quality and need to be reprocessed before publication. Below are some of my suggestions to be included in the final manuscript.

1. There are a lot of abbreviations in the manuscript and it would be commendable  if a list of abbreviations are prepared at the end of the article.

2. While performing method validation, the authors chose  only three points to correlate their method with LC-MS/MS(figure 4) which looks insufficient; at least 6 points should be tested to confirm the correlation(see reference 14 the work from your group).

3. the MS spectrum shown in the supplemntary information shows the need for further purification.

Author Response

Response: Thank you so much for your encouraging and helpful comments on our manuscript. Your support truly motivates us to keep pushing forward. The suggestions and issues you have raised are incredibly valuable to the article's improvement. We have carefully revised it line by line, highlighting the changes in red for your convenience during review. Once again, we sincerely appreciate your time and effort.

Reviewer 3 Report

Comments and Suggestions for Authors

Congratulation for the manuscript's scientific quality. Please take into consideration some improvements that are necessary to intervene the Introduction section.

Comments on the Quality of English Language

Needs minor language editing.

Author Response

(The authors gave the same response as above.)

Reviewer 4 Report

Comments and Suggestions for Authors

The authors present a paper that deals with the establishment of two methods for detecting Aflatoxin B1 in foodstuffs. These are based on the formation of a monoclonal antibody which is detected via  indirect competitive enzyme-linked immunosorbent assay (ic-ELISA) and time-resolved fluorescence immunoassay assay (TRFICA). I think this is an interesting study and could be published. I have a few comments on the current manuscript.

Abstract - the abstract is quite well written and summarises the method and main quantitative results from the two techniques

Introduction - 

I understand the authors point but feel this sentecne could be written more scientifically - " This causes it to accumulate in the animal body after consumption by animals [5], resulting in a cycle of pollution that encompasses the environment-feed-animal-food-human-environment con- 36 tinuum and ultimately harms human health and the ecological environment." 

Again I feel this could be better explained - "improve the sensitivity of strip"? "Meanwhile, due to the simplicity and convenience of LFIA, many efforts have been made in recent years to improve the sensitivity of strip, among which time- resolved fluorescence immunoassay assay (TRFICA) has emerged as a pivotal novel technique [11]."

Methodology - needs a bit more detail on the process of extraction and subsequent storage etc. "The reaction mixture was agitated at room temperature for 12 h, then was evap- 105 orated to dryness under N2 and redissolved in HCl (2mL, 0.05 mol/L), and ethylacetate (2 106 mL) was used to extract the AFB1-CMO hapten"

This part needs better explanation ". Subsequently, estimate the amount of hapten residue bound to the carrier molecule using the formula: (ε conjugation - ε protein) /ε hapten (ε: analyte absorbance coefficient)"

This part needs additional explanation/information "The specific detection procedure involves adding 100 μL AFB1 standard solution/sample to the sampling tank of the card, followed by incubation at 37°C for 6 min, and then qualitatively evaluating the results under an ultraviolet lamp."

Good that the authors compared their techniques with HPLC-MS/MS, but this part of the method description may need to be slightly expanded. Also this part could be expanded/written more clearly i.e. exact;y how was the comparison done, number of repeats etc. "The correlation between the results of HPLC-MS/MS, ic-ELISA, and TRFICA was calculated finally."

Results and discussion - Figure 1 is not that easy to intepret, can the authors highlight significant aspects better?

Figure 3 requires further explanation and a better figure legend with more detail. 

This part requires further explanation "AFB1 standard solution of 1 μg/L was used as a control, and ZEN, OTA, T-2 and DON standard solutions of 100 μg/L were respectively detected with the same batch of strip."  Also this seems like experimental description?

Needs a better description "As shown in Table 2, the recoveries of AFB1 added by ic-ELISA and TRFICA were both well," - both well?

Table 3 needs further discussion in the paper regarding the advances of this method. Or any differences with published studies. Also a discussion of future development of the assays to make them more commercially viable etc.

There also needs to be further discussion about the comparison to the GC-MS/MS method.

Conclusion - needs a summary of future directions for the work etc.

Comments on the Quality of English Language

English language is fine - there are just a few places in the manuscript where additional clarification, explanation is required.

Author Response

(The authors gave the same response as above.)

Round 2

Reviewer 2 Report

Comments and Suggestions for Authors

Minor corrections with regards to figure quality and font size  in the access still lacking. I hope it will be corrected before publication.

Author Response

Response: Thank you so much for your encouraging and helpful comments on our manuscript.  Your support truly motivates us to keep pushing forward.  The suggestions and issues you have raised are incredibly valuable to the article's improvement. We have revised the Figure 1, Figure 2 and Figure 4 of the manuscript to unify the font and size. Thank you very much for your careful review, which makes our article more rigorous and standardized. Once again, we sincerely appreciate your time and effort.

Reviewer 4 Report

Comments and Suggestions for Authors

The authors dealt with my comments and altered the manuscript o improve the clarity. I think the current manuscript is improved and could be published in Food.

apologies to the authors as in my original review I mentioned GC-MS/MS but meant to say there should be a better comparison with LC-MS/MS. I see that another reviewer also requested this comparison be emphasised/improved

Comments on the Quality of English Language

English language is fine

Author Response

The authors dealt with my comments and altered the manuscript o improve the clarity. I think the current manuscript is improved and could be published in Food.

Response: Thank you so much for your encouraging and helpful comments on our manuscript. Your support truly motivates us to keep pushing forward. The suggestions and issues you have raised are incredibly valuable to the article's improvement. Once again, we sincerely appreciate your time and effort.

apologies to the authors as in my original review I mentioned GC-MS/MS but meant to say there should be a better comparison with LC-MS/MS. I see that another reviewer also requested this comparison be emphasised/improved

Response: Thank you so much for pointing out the shortcomings of our manuscript. Chromatography is currently the gold standard method for AFB1 detection, so it is necessary to compare the method established in this article with chromatography and verify the correlation between methods. Therefore, in the method validation section (2.6) of this manuscript, we compared the ic-ELISA method with LC-MS/MS. The results showed that there was a great correlation between the methods, and the ic-ELISA established in this study was reliable. The comparison between ic-ELISA and TRFICA indirectly proved that the TRFICA method was reliable. And, we reconducted the validation experiment and increased the number to six points to correlate the methods. We replotted Figure 4. The linear regression analysis. (a) Correlation between TRFICA and ic-ELISA. (b) Correlation between ic-ELISA and LC-MS/MS. Meanwhile, for the LC-MS/MS method, we refer to the published articles (Biancardi & Dall’Asta, 2014) for experimental design to ensure the accuracy and reliability of the experiment. Thank you once again for your meticulous review, which greatly enhances the professionalism and academic rigor of our paper!
